# EXCOST: SEMI-SUPERVISED CLASSIFICATION WITH EXEMPLAR-CONTRASTIVE SELF-TRAINING

## ABSTRACT

Similar to the way of human learning, the aim of semi-supervised learning (SSL) method is to harness vast unlabeled data alongside a limited set of labeled samples. Inspired by theories of category representation in cognitive psychology, an innovative SSL algorithm named Exemplar-Contrastive Self-Training (EXCOST) is proposed in this paper. This algorithm ascertains pseudo-labels for unlabeled samples characterized by both substantial confidence and exemplar similarity, subsequently leveraging these pseudo-labels for self-training. Furthermore, a novel regularization term named Category-Invariant Loss (CIL) is applied for SSL. CIL promotes the generation of consistent class probabilities across different representations of the same sample under various perturbations, such as rotation or translation. Notably, the proposed approach does not depend on either the prevalent weak and strong data augmentation strategy or the use of exponential moving average (EMA). The efficacy of the proposed EXCOST is demonstrated through comprehensive evaluations on semi-supervised image classification tasks, where it attains state-of-the-art performance on benchmark datasets, including MNIST with 2, 5 and 10 labels per class, SVHN with 25 labels per class, and CIFAR-10 with 25 labels per class.

## 1 INTRODUCTION

Inspired by theories of category representation in cognitive psychology, we propose a semi-supervised learning (SSL) algorithm called Exemplar-Contrastive Self-Training (EXCOST). In cognitive psychology, two main theories explain how humans categorize: prototype theory (Rosch & Mervis, 1975; Rosch, 1975) and exemplar theory (Nosofsky, 1986; 1991), both extensively discussed in mainstream cognitive psychology textbooks (Anderson, 2020; Solso et al., 2007; Goldstein, 2018; Smith & Kosslyn, 2019). Prototype-based theories include central tendency theory and attribute-frequency theory. According to central tendency theory (Posner & Keele, 1968; Posner, 1969), prototypes are the average of features across a set of exemplars, an idea reflected in neural network models used for few-shot learning (Snell et al., 2017). On the other hand, attribute-frequency theory (Neumann, 1974; 1977; Solso & McCarthy, 1981) suggests that prototypes are combinations of the most frequent features. In contrast to prototype theory, exemplar theory emphasizes the distinctiveness of individual instances within a category, explaining how we make judgments about atypical cases. Although these seemingly opposing theories capture certain characteristics of human category representation, they are more complementary than contradictory: Prototypes prioritize commonalities and may overlook intra-class variability, while exemplars emphasize individuality at the risk of sacrificing generalization abilities. While early research aimed to determine whether categories are represented by exemplars or prototypes, it is more reasonable to consider that both representations are available. Some categories may be better represented by exemplars, while others may be better represented by prototypes (Smith, 2014).

Self-training (Yarowsky, 1995) is a SSL approach aimed at leveraging unlabeled data to improve the performance of supervised learning models. This method assumes the availability of a certain amount of well-labeled data for training. However, due to various reasons, acquiring a sufficient amount of labeled data is challenging. In such scenarios, leveraging predictions on unlabeled data can assist in training an improved model, thereby enhancing performance and reducing reliance on labeled data. However, this approach also has certain limitations. For instance, prediction errors can lead to incorrect labels, subsequently affecting the model's performance. This phenomenon is

known as confirmation bias (Tarvainen & Valpola, 2017; Arazo et al., 2020). Confirmation bias refers to the tendency of individuals to favor information that aligns with their existing beliefs while disregarding information that contradicts them. In the context of models with limited predictive capabilities, there is a risk of making erroneous predictions on unlabeled samples and using these mislabeled samples for subsequent training, thus reinforcing this bias.

Based on the aforementioned observations, we propose a SSL algorithm called Exemplar-Contrastive Self-Training (EXCOST). It combines prototype and exemplar information to provide higher-quality pseudo-labels during the self-training process. EXCOST consistently maintains highly accurate predictions for unlabeled data, thus mitigating the issue of confirmation bias to a significant extent. Notably, the proposed model achieves performance comparable to state-of-the-art supervised learning models on the MNIST dataset, requiring only two labeled samples per class.

The notion of typicality gradient plays a fundamental role in our understanding of category processing (Smith & Kosslyn, 2019). It highlights the continuous nature of the similarity relationship between members and category prototypes within a given category. Typicality gradients are observed across various cognitive domains, including language, perception, and memory. For instance, typical category members (e.g., "apple" for the category of fruits) elicit faster and more accurate responses in classification tasks compared to atypical ones (e.g., "durian") (Rosch, 1975; Posner & Keele, 1968). The presence of typicality gradients in category processing has important implications for our understanding of how categories are represented and classified. It suggests that categories are not simply defined by a fixed set of features or a binary categorization, but rather exhibit a graded structure where some members are more representative of the category than others. This graded structure enables us to make fine-grained distinctions among entities and respond to category-related stimuli in a nuanced manner.

Building upon the research findings mentioned above, we introduce the Category-Invariant Loss (CIL) as a novel regularization term. We emphasize the preservation of invariance and typicality gradients at the category level. Invariance plays a crucial role in the principles of contrastive learning and consistency regularization, and the main objective of the CIL is to emphasize the presence of typicality gradients at the abstract level of categories, while ensuring that these gradients remain invariant across different views of the same instance.

In summary, our main contributions are as follows:

- The proposal of Exemplar-Contrastive Self-Training (EXCOST), a SSL algorithm that generates more reliable pseudo-labels by combining measures of confidence and exemplar similarity.

- The proposal of Category-Invariant Loss (CIL), a regularization term that does not require the support of pseudo-labels and encourages the model to produce similar class probabilities for the same sample under various perturbations.

- The proposed algorithm was empirically evaluated on semi-supervised image classification tasks using the MNIST, CIFAR-10/100, and SVHN datasets. The results demonstrated the proposed algorithm achieves state-of-the-art performance on MNIST with 2, 5, and 10 labels per class, as well as on SVHN and CIFAR-10 with 25 labels per class.

It is worth noting that the proposed method did not employ the recent popular weak and strong data augmentation strategy (Sohn et al., 2020; Berthelot et al., 2020; Zhang et al., 2021; Wang et al., 2023), nor did it utilize exponential moving average (EMA) (Tarvainen & Valpola, 2017; Sohn et al., 2020; Zhang et al., 2021; Wang et al., 2023).

## 2 RELATED WORK

Self-Training (Yarowsky, 1995) is a prominent technique in the field of semi-supervised learning (SSL). The core idea behind Self-Training involves training a model using labeled data and then exploiting this model to predict labels for unlabeled data instances. These predicted labels, referred to as pseudo-labels, are treated as ground truth during the subsequent training iterations. This iterative process is repeated until convergence is achieved. Self-Training has garnered significant attention and yielded impressive results across various domains, including image classification (Lee, 2013;

Xie et al., 2020b), object detection (Rosenberg et al., 2005), and natural language processing (Mc-Closky et al., 2006).

Consistency regularization (Bachman et al., 2014) plays a crucial role in SSL by promoting robustness and generalization. By encouraging consistent predictions under perturbations, it helps the model capture meaningful patterns and reduce sensitivity to noisy variations. The use of data augmentation techniques, such as random rotations, translations, and flips, or adversarial perturbation, introduces diverse views of the same samples during training (Sajjadi et al., 2016; Samuli & Timo, 2017; Tarvainen & Valpola, 2017; Berthelot et al., 2019; Miyato et al., 2019; Berthelot et al., 2020; Sohn et al., 2020; Xie et al., 2020a). This increases the model's exposure to different variations of the data and enhances its ability to generalize to unseen samples. Similarly, model perturbations are also employed to maintain consistency. Dropout (Srivastava et al., 2014), as a widely used regularization technique, randomly masks out neurons during training, forcing the model to adapt to different subsets of the network (Sajjadi et al., 2016; Samuli & Timo, 2017; Tarvainen & Valpola, 2017). Alternatively, applies random max-pooling operations to the input data, introducing spatial perturbations (Sajjadi et al., 2016). Furthermore, ensembling models trained at different epochs effectively creates an ensemble of models with different training dynamics (Samuli & Timo, 2017; Tarvainen & Valpola, 2017). These perturbations, whether applied to samples or models, aim to enrich the training process by injecting diverse sources of information and inducing a form of regularization that enhances the model's robustness.

Contrastive learning has emerged as a powerful paradigm for learning effective representations in few-shot learning. The core idea behind contrastive learning is to train a model to distinguish between positive and negative pairs, thus encouraging the model to capture meaningful patterns and similarities. Tasks such as signature verification (Bromley et al., 1993), face verification (Chopra et al., 2005; Taigman et al., 2014), and one-shot image recognition (Koch et al., 2015) have benefited from the principles of contrastive learning. By leveraging contrastive learning, these tasks can effectively learn discriminative representations and achieve promising performance, even with limited data available. Moreover, contrastive learning has emerged as a popular paradigm in the realm of self-supervised learning (He et al., 2020; Chen et al., 2020; Grill et al., 2020; Caron et al., 2020; Chen & He, 2021). By formulating pretext tasks that require models to capture meaningful patterns or relationships within the data, contrastive learning enables unsupervised representation learning. Furthermore, the concept of contrastive learning has been extended to multimodal learning (Radford et al., 2021), where it enables effective fusion of information from different modalities. By employing contrastive learning principles, multimodal models can learn joint representations that capture relevant cross-modal interactions, ultimately enhancing the understanding and utilization of multimodal data.

## 3 THE PROPOSED METHOD

### 3.1 CATEGORY-INVARIANT LOSS

In deep neural networks, it has been observed that feature vectors in the last hidden layer exhibit significant semantic similarity (Krizhevsky et al., 2012; Wu et al., 2018). Different visual perspectives of the same object should have similar feature vector representations, upon which the classification head can generate category representations robust to viewpoint changes. This observation aligns with findings in neuroscience, where researchers often refer to this phenomenon as perceptual constancy (Kandel et al., 2021), which denotes the ability to recognize objects stably despite variations in viewpoint or lighting conditions. Similar to the human visual system, in which higher cortical areas process more abstract information and exhibit greater invariance (Freedman et al., 2002). It is worth noting that demanding invariance in feature vectors can potentially threaten the model's performance, as we will elucidate in Appendix D.

Furthermore, considering the characteristics of neural networks based on backpropagation, it is known that weights of connections get strengthened when high-frequency features co-occur with specific class labels. Thus, to some extent, category learning in neural networks aligns with the prototype theory based on attribute frequency. As prototype theory emphasizes the existence of typicality gradients, when predictions of the network are regarded as a representation of categories, we expect it to retain a form of typicality gradient. In the context of semi-supervised learning with sparse labeled data, due to the lack of comprehensive class labels, the network's predictions

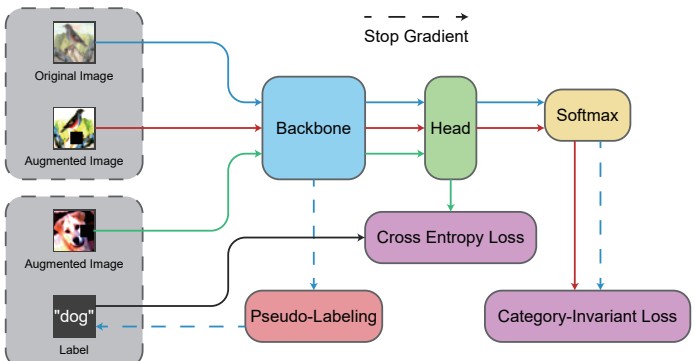

Figure 1: An overview of the EXCOST algorithm is presented. Both original (blue lines) and augmented images (red lines) are input to the backbone for feature extraction. These features then go to the classification head, where the Softmax function transforms them, resulting in class probabilities. The Category-Invariant Loss (CIL) is computed in this process, with no gradient passing through the pathway of original images. The generated feature vectors for original images are kept for generating pseudo-labels used for the next epoch. Images with labels (green lines) use cross-entropy loss for optimization.

inherently carry gradients. In other words, the network exhibits varying levels of confidence in predicting different data samples. However, popular models for semi-supervised learning using "hard" pseudo-label optimization may lead to the disappearance of typicality gradients. This is because strict assignment of samples to discrete categories essentially lacks the concept of typicality. In the absence of guaranteed label quality, this may result in confirmation bias appearing early in training.

Based on the aforementioned requirements regarding invariance and typicality gradients, we propose the Category-Invariant Loss (CIL), specifically, the formula for CIL is as follows:

$$\mathcal{L}_u = \frac{1}{B_u} \sum_{i=1}^{B_u} \max\left(\frac{\max(f(\boldsymbol{x}_i; \tilde{\boldsymbol{\theta}})) - \psi}{1 - \psi}, 0\right) \cdot \left(1 - \frac{f(\boldsymbol{x}_i; \tilde{\boldsymbol{\theta}})}{\|f(\boldsymbol{x}_i; \tilde{\boldsymbol{\theta}})\|} \cdot \frac{f(T(\boldsymbol{x}_i); \boldsymbol{\theta})}{\|f(T(\boldsymbol{x}_i); \boldsymbol{\theta})\|}\right) \quad (1)$$

where $\psi$ is a threshold, $f(\boldsymbol{x}; \boldsymbol{\theta})$ represents the model's predicted class probabilities for the sample $\boldsymbol{x}$, with $\boldsymbol{\theta}$ denoting the model's trainable parameters. $\tilde{\boldsymbol{\theta}}$ does not participate in gradient backpropagation and serves as a fixed replica of $\boldsymbol{\theta}$. $T(\cdot)$ denotes the data augmentation function, and $B_u$ is the batch size.

It is noteworthy that CIL shares a striking resemblance to the negative cosine similarity utilized in self-supervised contrastive learning (Chen & He, 2021). The primary distinction lies in CIL's emphasis on achieving semantic-level abstraction invariance within the context of SSL tasks, thus operating on class probabilities. Moreover, to alleviate the influence of irrelevant or outlier samples, we introduce a threshold. During the early stages of model training, a relatively lower threshold is employed to expose the model to a broader range of samples. As training progresses, the model tends to assign higher confidence to samples, necessitating a higher threshold to safeguard against the impact of outlier samples. Samples' confidence falling below the threshold contribute no loss, while samples' confidence surpassing the threshold have their loss weights adjusted based on their confidence. This adaptive weighting scheme enables the model to prioritize highly confident samples, granting them greater influence throughout the learning process.

By incorporating CIL into the training process, we expect the model to learn more robust and semantically consistent representations, effectively addressing the limitations imposed by excessive invariance demands and potential confirmation bias caused by "hard" pseudo-labels. This novel loss function promotes the preservation of the typicality gradient while ensuring category invariance across different views of the same instance.

## 3.2 EXEMPLAR-CONTRASTIVE SELF-TRAINING

In the context of SSL tasks, we define $\mathbb{D} = \{(\boldsymbol{x}_i, \boldsymbol{y}_i)\}_{i=1}^{N_S}$ as the dataset, where $\boldsymbol{x}_i$ represents the $i$-th sample within the dataset $\mathbb{D}$, while $\boldsymbol{y}_i$ denotes its associated label. The label, $\boldsymbol{y}_i$, is an integer value ranging from $-1$ to $C-1$, with $C$ denoting the total number of classes. Samples with a label of $\boldsymbol{y}_i = -1$ means that the corresponding sample $\boldsymbol{x}_i$ has not yet been labeled. The total number of samples is denoted by $N_S$. Furthermore, we define the original label sequence as $\boldsymbol{l}_O$. Due to the fact that only the samples associated with non-"-1" labels in $\boldsymbol{l}_O$ provide category-related information, we refer to these samples as "exemplars" to distinguish them from the remaining samples.

According to exemplar theory, instances belonging to the same category should exhibit similarity. Therefore, we need to compute the similarity between samples and exemplars. Given that the output of the classification head provides category probabilities but lacks fine-grained details, we compare the feature vectors output from the backbone network. Let $\boldsymbol{x}$ denote samples, and $\boldsymbol{e}$ represent exemplars. We use cosine similarity to compute the similarity scores $\boldsymbol{S}$ between $\boldsymbol{x}$ and $\boldsymbol{e}$:

$$\boldsymbol{S}_{i,j} = \frac{h(\boldsymbol{x}_i; \boldsymbol{\theta})}{\|h(\boldsymbol{x}_i; \boldsymbol{\theta})\|} \cdot \frac{h(\boldsymbol{e}_j; \boldsymbol{\theta})}{\|h(\boldsymbol{e}_j; \boldsymbol{\theta})\|} \tag{2}$$

where $\boldsymbol{S}_{i,j}$ denotes the similarity between the $i$-th sample and the $j$-th exemplar, $h(\cdot; \boldsymbol{\theta})$ denotes the feature vector output from the backbone network. Each epoch of EXCOST consists of two sequential phases: training and pseudo-label generation. During the training phase, we store the values of $h(\boldsymbol{x}; \boldsymbol{\theta})$. Given the relatively limited number of exemplars, the computation of $h(\boldsymbol{e}; \boldsymbol{\theta})$ is deferred until the labeling phase, ensuring a more up-to-date output of the network. By employing this strategy, we effectively mitigate the overall computational burden, thus enhancing training efficiency.

In addition, we employ a classification head to generate class probabilities for predictions on $h(\boldsymbol{x}; \boldsymbol{\theta})$. These class probabilities are referred to as $\boldsymbol{P}$, with $\boldsymbol{P}_{i,k}$ representing the model's prediction probability that the $i$-th sample belongs to the $k$-th class. This approach aligns with the prototype theory based on attribute frequencies. In the context of classification tasks, the optimization process of deep neural networks involves establishing associations between features and class labels, wherein features with higher frequencies form stronger connections with their corresponding classes. Indeed, employing high-confidence model predictions as pseudo-labels has been widely adopted (Sohn et al., 2020; Zhang et al., 2021; Rizve et al., 2021; Wang et al., 2023). Therefore, we also underscore the significance of exemplars. By integrating prototype theory and exemplar theory, we capitalize on the strengths of both approaches, resulting in a robust framework for pseudo-label generation. Prototype theory explains how we capture the essential characteristics of distinct categories. On the other hand, exemplar theory elucidates how we harness the rich information inherent in the data distribution.

Throughout the entire training process, we gradually assign labels to unlabeled samples based on $\boldsymbol{S}$ and $\boldsymbol{P}$. To adjust the labeling rate, we employ labeling rate functions that are contingent on the training progress, denoted as $\Phi_S$ (based on sample similarity) and $\Phi_C$ (based on confidence). As training progresses, $\Phi_S$ and $\Phi_C$ increase gradually, allowing the model to incorporate more pseudo-labeled samples into the learning process.

As described in Algorithm 1, EXCOST selects samples that simultaneously satisfy the following conditions for labeling: (1) The samples have a confidence ranking within the top $k_P$ as predicted by the model for that class. Here, $k_P$ is calculated by multiplying $\phi_C$ with the estimation of the number of samples belonging to that class. Due to the unknown sample distribution, we obtain this estimation by applying a weighted average using $\phi_C$ for the class-specific prediction count and the total number of samples divided by the number of classes. During the early stages of training, when the model is still immature and lacks substantial information, a small value of $\phi_C$ implies fewer labels need to be generated. Under this circumstance, considering a uniform distribution of data is reasonable. In contrast, as training progresses, $\phi_C$ generally becomes larger, indicating the model's maturity. Consequently, we believe that the model's judgment about the data distribution of the class is more accurate. Therefore, this estimation gradually transforms from a uniform distribution to the model's predicted class marginal distribution as $\phi_C$ increases. (2) The feature vector of a sample exhibits the highest similarity with the feature vector of a specific exemplar belonging to the same class as predicted by the model, with the similarity score ranking among the top $k_S$. As the model

---

**Algorithm 1** EXCOST algorithm during labeling phase

---

**Input:** number of classes $C$, number of samples $N_S$, original sequence of labels $\boldsymbol{l}_O$, similarity scores matrix $\boldsymbol{S}$, class probabilities matrix $\boldsymbol{P}$, margin $\delta$, labeling rate based on confidence $\phi_C$, labeling rate based on exemplar similarity $\phi_S$.

$\boldsymbol{l}_P \leftarrow \boldsymbol{l}_O$ {Initialize sequence of labels $\boldsymbol{l}_P$}

$\xi_E \leftarrow \text{argwhere}(\boldsymbol{l}_O \neq -1)$ {Find the indices of exemplars}

$\boldsymbol{p}_V, \boldsymbol{p}_I \leftarrow \max(\boldsymbol{P})$ {Find the maximum value and index for each sample in the $\boldsymbol{P}$}

$\boldsymbol{s}_V, \boldsymbol{s}_I \leftarrow \max(\boldsymbol{S})$ {Find the maximum value and index for each sample in the $\boldsymbol{S}$}

**if** $\phi_C > 0$ **and** $\phi_S > 0$ **then**
    **for** $i = 0$ to $C - 1$ **do**
        $\xi_P \leftarrow \text{argwhere}(\boldsymbol{p}_I = i)$
        **if** $\text{length}(\xi_P) > 0$ **then**
            $k_P \leftarrow \min(\text{length}(\xi_P), \text{ceil}(\phi_C \cdot (\phi_C \cdot \text{length}(\xi_P) + (1 - \phi_C) \cdot \frac{N_S}{C})))$
            $\xi_P \leftarrow \xi_P[\boldsymbol{p}_V[\xi_P] \geq \max(1 - \delta, \min(\text{topk}(\boldsymbol{p}_V[\xi_P], k_P)))]$ {Update $\xi_P$ by selecting indices with values greater than both $1 - \delta$ and the $k_P$-th largest confidence}
            $\xi_{E,i} \leftarrow \text{argwhere}(\boldsymbol{l}_O[\xi_E] = i)$ {Find the indices of exemplars that belong to class $i$}
            **for** $j = 0$ to $\text{length}(\xi_{E,i})$ **do**
                $\xi_S \leftarrow \text{argwhere}(\boldsymbol{s}_I = \xi_{E,i}[j])$
                **if** $\text{length}(\xi_S) > 0$ **then**
                    $k_S \leftarrow \phi_S \cdot \text{length}(\xi_S)$
                    $\xi_S \leftarrow \xi_S[\boldsymbol{s}_V[\xi_S] \geq \min(\text{topk}(\boldsymbol{s}_V[\xi_S], k_S))]$ {Update $\xi_S$ by selecting indices with values greater than the $k_S$-th largest similarity score}
                    $\boldsymbol{l}_P[(\{\xi_P\} \cap \{\xi_S\}) \backslash \{\xi_E\}] \leftarrow i$
                **end if**
            **end for**
        **end if**
    **end for**
**end if**
**Return:** $\boldsymbol{l}_P$

---

comprehensively learns from exemplars right from the outset, and each exemplar holds distinct characteristics, there exists no justification to assume that data would naturally partition uniformly based on exemplars. Therefore, $k_S$ is directly obtained by multiplying $\phi_S$ with the number of samples that are most similar to that specific exemplar. (3) In order to ensure the avoidance of selecting samples with excessively low confidence during the early stages of model training when it is less mature, and to prevent the model from choosing outlier samples that could disrupt stability in the later stages of training when $\phi_C$ is significantly larger, we introduce a additional hyperparameter, referred to as "margin," denoted by $\delta$. This hyperparameter further constrains the confidence values within the interval range of $[1 - \delta, 1]$. We will demonstrate the accuracy of the EXCOST algorithm in generating pseudo-labels in Appendix E.

During the training phase, we treat labels indistinguishably, whether they are ground truth labels or pseudo-labels. During the initial epoch, only the ground truth labels are utilized as the pseudo-label generation process has not yet commenced. In the training phase, we need to store feature vectors for all samples. To ensure that each sample passes through the backbone network at least once, we must evenly distribute all training samples across different iterations of each epoch. For labeled samples, we ensure an even distribution across iterations for each class. This practice serves to maintain balance among classes, preventing the model from exhibiting bias towards any specific class. We employ cross-entropy loss to optimize the proposed model using the labels $\boldsymbol{l}_P$ along with their corresponding samples:

$$\mathcal{L}_s = \frac{1}{B_s} \sum_{i=1}^{B_s} - \log \frac{e^{f(\boldsymbol{x}_i; \boldsymbol{\theta})_{\boldsymbol{y}_i}}}{\sum_{j=0}^{C-1} e^{f(\boldsymbol{x}_i; \boldsymbol{\theta})_j}} \tag{3}$$

where $B_s$ is the batch size. Finally, the overall loss of EXCOST results from a weighted combination (using $w_u$) of the CIL loss, denoted as $\mathcal{L}_u$, which is applied to the entire set of samples, and the cross-entropy loss, denoted as $\mathcal{L}_s$, which is computed from the labeled samples:

$$\mathcal{L} = w_u \mathcal{L}_u + \mathcal{L}_s \tag{4}$$

## 4 EXPERIMENTS

### 4.1 SETUP

We evaluated EXCOST on the following datasets: MNIST (LeCun et al., 1998), CIFAR-10/100 (Krizhevsky, 2009), and SVHN (Netzer et al., 2011). Before 2019, MNIST stood as a pivotal dataset in the field of semi-supervised image classification. Despite researchers gradually shifting towards more challenging datasets for image classification tasks in recent years, MNIST remains a significant benchmark dataset. Another reason for including MNIST is that, to the best of our knowledge, no prior studies have reported semi-supervised learning results that match supervised learning performance across all these datasets simultaneously. We conducted experiments with various quantities of labeled samples commonly used in previous research for each dataset.

Within the EXCOST algorithm, several hyperparameters require consideration: the total number of training epochs $T$, the number of training iterations per epoch $K$, the number of labeled samples per class per iteration $N_C$, the threshold scheduling function for CIL denoted as $\Psi$, the scheduling function for labeling rate based on confidence denoted as $\Phi_C$, the scheduling function for labeling rate based on exemplar similarity denoted as $\Phi_S$, and the margin $\delta$ ensuring stable labeling. The batch size $B_u$ for CIL does not require an additional setting, as it is determined by $\lfloor \frac{N_S}{K} \rfloor$, ensuring that all samples pass through the backbone network once in each epoch. Consequently, the total number of samples $N_S$ must be divisible by $K$. Similarly, the batch size $B_s$ in the cross-entropy loss is also not set separately, it is obtained from $N_C \cdot C$. For $T$, we set it to 200 for both MNIST and SVHN, while for CIFAR-10/100, we set it to 600. As for $K$, the values are 75 for MNIST, 732 for SVHN, and 500 for CIFAR-10/100. Please note that, as the total number of samples in SVHN is 73257, the value of $B_u$ calculated above is 100. Therefore, we have utilized only 73200 samples. For $N_C$, we directly compute it using $\frac{N_S}{K \cdot C}$, resulting in $B_u$ being equal to $B_s$ in this scenario. The parameter $\delta$ is set uniformly to 0.01 across all datasets. $\Psi$, $\Phi_C$, and $\Phi_S$ are constructed using the sigmoid function and are adjusted through truncation, translation, and scaling operations, all of which can be expressed in the following form:

$$\sigma(t; \alpha_0, \alpha_1, \beta_0, \beta_1, \gamma) = \min\left(\max\left(\frac{\alpha_1}{1 + e^{-\frac{t \cdot (\beta_1 - \beta_0) + \beta_0}{\alpha_0}}} + \gamma, 0\right), 1\right) \tag{5}$$

The parameters $\alpha_0$ and $\alpha_1$ control the scaling factors of the sigmoid function along the X and Y axes respectively, while $\beta_0$ and $\beta_1$ set the starting and ending points of the function along the X-axis. The parameter $\gamma$ enables the function to be shifted along the Y-axis. The variable $t$ represents the current training progress, with a range of $(0, 1]$. For the function $\Psi$, we utilize the following parameter values across all datasets: $(\alpha_0 = \frac{10}{T}, \alpha_1 = 2 \cdot (0.9 - \frac{1}{C}), \beta_0 = 0, \beta_1 = 1, \gamma = -(0.9 - \frac{2}{C}))$. This choice gradually increases the threshold from $\frac{1}{C}$ to approximately 0.9, reaching its peak value after around 80 epochs of training. As for the function $\Phi_C$, we adopt different parameter values for different datasets. For MNIST and CIFAR-10/100, we use $(\alpha_0 = 0.3, \alpha_1 = 0.99, \beta_0 = -1, \beta_1 = 2, \gamma = 0)$, while for SVHN, the parameters are $(\alpha_0 = 0.5, \alpha_1 = 0.99, \beta_0 = -1, \beta_1 = 4, \gamma = 0)$. For the function $\Phi_S$, we set it as a sigmoid function with parameters identical to those of $\Phi_C$ in the MNIST and SVHN datasets. However, in the case of CIFAR-10/100, due to the greater diversity of samples, we observed better performance when it was set as a constant function with a value of 1.

With the exception of CIFAR-100, we employed the Adam optimization algorithm (Kingma & Ba, 2017) for the remaining datasets. We chose the default recommended parameters, specifically $(\alpha = 0.001, \beta_1 = 0.9, \beta_2 = 0.999, \epsilon = 10^{-8})$. We initiated a linear decrease in the learning rate once the training progress reached 60%, gradually reducing it to $\frac{1}{10}$ of the initial value. However, for CIFAR-100, we observed that achieving satisfactory results with the Adam optimization algorithm in this context was challenging. As a result, we opted for the SGD optimization algorithm with a learning rate of 0.03 and a momentum of 0.9. Additionally, a weight decay of $10^{-5}$ was applied. The learning rate is decreased to 30% of its previous value at every 10% training progress interval. In all cases, we set $w_u$ to 1, except for the CIFAR-100, where we set $w_u$ to 10.

Table 1: Error rates on MNIST, CIFAR-10/100, and SVHN datasets. Methods annotated with † indicate that they report the median error rate of the last 20 epochs, while methods annotated with ‡ indicate that they report the minimum error rate among all epochs. Bold type highlights the minimum value in that column, and an underline means a value that is not greater than 110% of the minimum value. Due to the additional 531,131 samples in the SVHN dataset, we present separately the methods that utilize this extra data.

| Dataset | MNIST | | | CIFAR-10 | | CIFAR-100 | | SVHN | | SVHN+Extra | |
|---|---|---|---|---|---|---|---|---|---|---|---|
| Labels | 20 | 50 | 100 | 250 | 4000 | 2500 | 10000 | 250 | 1000 | 250 | 1000 |
| LadderNetwork (Rasmus et al., 2015) | | | 0.89±0.50 | | 20.40±0.47 | | | | | | |
| ImprovedGAN (Salimans et al., 2016) | 11.34±4.45 | 1.42±0.96 | 0.86±0.06 | | 15.59±0.47 | | | | 5.88±1.00 | | |
| SNTG (Luo et al., 2018) | 1.36±0.78 | 0.94±0.42 | 0.66±0.07 | | 9.89±0.34 | | 37.97±0.29 | 4.29±0.23 | 3.82±0.25 | | |
| MixMatch† (Berthelot et al., 2019) | | | | | 4.95±0.08 | | 25.88±0.30 | 3.78±0.26 | 3.27±0.31 | 2.22±0.08 | 2.18±0.06 |
| FixMatch† (Sohn et al., 2020) | | | | 5.07±0.65 | 4.26±0.05 | 28.29±0.11 | 22.60±0.12 | 2.48±0.38 | 2.28±0.11 | | |
| UDA (Xie et al., 2020a) | | | | | 4.32±0.08 | | | | 2.23±0.07 | | |
| ReMixMatch† (Berthelot et al., 2020) | | | | 6.27±0.34 | 5.14±0.04 | | | 3.10±0.50 | 2.83±0.30 | | |
| MPL (Pham et al., 2021) | | | | | 3.89±0.08 | | | | | | 1.99±0.07 |
| FlexMatch‡ (Zhang et al., 2021) | | | | 4.98±0.09 | 4.19±0.01 | 26.49±0.20 | 21.90±0.15 | | | | 6.72±0.30 |
| NP-Match (Wang et al., 2022) | | | | 4.96±0.06 | 4.11±0.02 | 26.03±0.26 | 21.22±0.13 | | | | |
| FreeMatch‡ (Wang et al., 2023) | | | | 4.88±0.18 | 4.10±0.02 | 26.47±0.20 | 21.68±0.03 | | | 1.97±0.01 | 1.96±0.03 |
| EXCOST (minimum of all epochs) | **0.32±0.02** | **0.34±0.04** | **0.32±0.02** | 4.56±0.18 | 4.07±0.04 | 35.67±0.80 | 25.88±0.14 | **2.36±0.03** | 2.33±0.06 | | |
| EXCOST (median of last 20 epochs) | 0.39±0.06 | 0.40±0.05 | 0.40±0.03 | 4.74±0.16 | 4.28±0.04 | 35.95±0.70 | 26.23±0.12 | 2.50±0.05 | 2.44±0.03 | | |

For the MNIST dataset, we employed a network architecture consisting of 5 convolutional layers followed by 2 fully connected layers. In the case of SVHN and CIFAR-10/100, we adopted a variant of wide residual networks (WRNs) (Zagoruyko & Komodakis, 2016). This variant employed the Mish activation function (Misra, 2019) and allowed for an adjustable number of convolutional kernels in the first layer, as opposed to a fixed 16. Moreover, we permitted the use of global max pooling in lieu of global average pooling. Specifically, for SVHN, we employed the WRN-28-2 architecture. However, the first convolutional layer utilized 64 kernels, resulting in a significantly increased number of parameters compared to the original version; moreover, global max pooling was employed. On the other hand, for both CIFAR-10 and CIFAR-100, we implemented a WRN-28-8 architecture. Notably, the distinguishing factor between them lay in the number of kernels in their first convolutional layers. CIFAR-10 employed 16 kernels, while CIFAR-100 utilized 32; both employed global average pooling.

For the MNIST dataset, we employed four types of random transformations, namely rotation, translation, scaling, and shearing, as data augmentation strategy. For SVHN, we opted for random affine transformations. Interestingly, we observed that a single random affine transformation performed on the SVHN dataset consistently outperformed applying the aforementioned four spatial transformations individually. Additionally, we introduce various random color adjustments and incorporate random erasing (Zhong et al., 2020). For the CIFAR-10 and CIFAR-100 datasets, we adopt an identical set for data augmentation, encompassing the four aforementioned spatial transformations, random color adjustments, and random erasing. Please refer to Appendix A for more details.

## 4.2 RESULTS

We present the performance of all baselines and EXCOST in Table 1. The error rates reported for the baseline methods are taken from the original papers. For EXCOST, we conducted experiments using different random seeds and reported the results as averages. For the MNIST dataset, we performed 10 runs. For CIFAR-10/100 and SVHN, we conducted 3 runs. These results demonstrate that EXCOST achieves state-of-the-art performance on MNIST with 2, 5 and 10 labels per class, SVHN with 25 labels per class, and CIFAR-10 with 25 labels per class. Additionally, we analyze the labels produced by EXCOST on MNIST that are inconsistent with the dataset's ground truth labels. It is noteworthy that most of these inconsistencies cannot be attributed to erroneous EXCOST labeling; rather, they mainly stem from ambiguously defined samples and human labeling errors. Further details are presented in Appendix B. Due to limited hardware resources, we encountered challenges in identifying a sufficiently satisfactory set of hyperparameters for CIFAR-100. It's worth noting that in optimal conditions, there's a possibility that EXCOST could demonstrate improved performance on CIFAR-100.

### 4.3 ABLATION STUDY

EXCOST consists primarily of two key components: CIL and the labeling algorithm of EXCOST. For ease of exposition, we define the following experimental settings:

- $\mathcal{S}_1$: Employing CIL with $\Psi$ being a sigmoid function parameterized as ($\alpha_0 = \frac{10}{T}, \alpha_1 = 2 \cdot (0.9 - \frac{1}{C}), \beta_0 = 0, \beta_1 = 1, \gamma = -(0.9 - \frac{2}{C})$).
- $\mathcal{S}_2$: Employing CIL with $\Psi$ being a constant function with a value of 0.
- $\mathcal{S}_3$: Considering exemplar similarity in the labeling algorithm, with both $\Phi_C$ and $\Phi_S$ being sigmoid functions parameterized as ($\alpha_0 = 0.3, \alpha_1 = 0.9, \beta_0 = -1, \beta_1 = 2, \gamma = 0$).
- $\mathcal{S}_4$: Considering exemplar similarity in the labeling algorithm, with both $\Phi_C$ and $\Phi_S$ being sigmoid functions parameterized as ($\alpha_0 = 0.3, \alpha_1 = 0.99, \beta_0 = -1, \beta_1 = 2, \gamma = 0$).
- $\mathcal{S}_5$: Considering exemplar similarity in the labeling algorithm, with both $\Phi_C$ and $\Phi_S$ being sigmoid functions parameterized as ($\alpha_0 = 0.3, \alpha_1 = 1.0, \beta_0 = -1, \beta_1 = 2, \gamma = 0$).
- $\mathcal{S}_6$: Labeling algorithm with a margin of $0.01$.
- $\mathcal{S}_7$: Labeling algorithm with a margin of $1.0$.
- $\mathcal{S}_8$: Not using CIL.
- $\mathcal{S}_9$: Not considering exemplar similarity in the labeling algorithm.
- $\mathcal{S}_{10}$: Considering exemplar similarity in the labeling algorithm with $\Phi_C$ being a sigmoid function parameterized as ($\alpha_0 = 0.3, \alpha_1 = 0.99, \beta_0 = -1, \beta_1 = 2, \gamma = 0$), and $\Phi_S$ being a constant function with a value of $1$.

Given the various combinations of experimental settings, conducting comprehensive ablation experiments on complex datasets can be challenging. Therefore, we initially conducted experiments on the MNIST dataset with 20 labels. $\mathcal{S}1 \ldots \mathcal{S}7$ can be partitioned into three sets, resulting in 12 different experimental setting combinations by selecting one element from each subset. Surprisingly, we found that the proposed model is not sensitive to these parameters across all 12 different experimental setting combinations. The lowest error rate was $0.324\%$, the highest was only $0.505\%$, and the average error rate was $0.394\%$. When $\mathcal{S}_9$ was added, the average error rate increased to $0.402\%$. When we replaced $\mathcal{S}_1$ and $\mathcal{S}_2$ with $\mathcal{S}_8$, the average error rate rose to $1.636\%$, and further adding $\mathcal{S}_9$ resulted in an average error rate of $1.748\%$. Therefore, it is evident that satisfactory results were achieved using only CIL in the semi-supervised task on the MNIST dataset with 20 labels.

The experiments mentioned above suggest that the role of exemplar similarity may not be significant, but experiments on other datasets have confirmed the importance of exemplar similarity. For instance, we conducted ablation experiments on the more challenging CIFAR-10 dataset with 250 labeled samples, to investigate the impact of exemplar similarity in the labeling function. The error rate was $4.933\%$ when using experimental settings $\{\mathcal{S}_1, \mathcal{S}_4, \mathcal{S}_6\}$. After including $\mathcal{S}_9$, the error rate increased to $4.967\%$. However, when we used experimental settings $\{\mathcal{S}_1, \mathcal{S}_{10}, \mathcal{S}_6\}$, the error rate decreased to $4.560\%$. Clearly, for this task, a loose constraint on exemplar similarity is necessary to achieve optimal results, indicating that exemplar similarity remains crucial. Please refer to Appendix C for detailed experimental particulars.

## 5 CONCLUSIONS

Inspired by cognitive psychology, we proposed the EXCOST algorithm. EXCOST only requires data augmentation suitable for the dataset without being restricted to specific forms of data augmentation, thus demonstrating its wide applicability. The excellent performance of EXCOST on both MNIST and CIFAR-10, two datasets with significant differences, further underscores this point. The proposed approach achieves state-of-the-art or highly competitive results on MNIST, SVHN, and CIFAR-10. It is worth noting that these results were obtained without the use of EMA. The further discussion about the superiority of the proposed algorithm can be found at Appendix F.

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

# A   DETAILS OF DATA AUGMENTATION

Table 2: List of data augmentations employed in our experiments.

| Transformation | Description |
|---|---|
| RandomRotation($t_0$, $t_1$, $p$) | With a probability of $p$, perform a random rotation within the range of $[t_0, t_1]$ degrees. |
| RandomTranslation($x$, $y$, $p$) | With a probability of $p$, perform a random translation within the horizontal range of $[-x, x]$ and the vertical range of $[-y, y]$. |
| RandomScaling($t_0$, $t_1$, $p$) | With a probability of $p$, perform a random scaling within the range of $[t_0, t_1]$, where a value of 1 means no scaling. |
| RandomShear($x_0$, $x_1$, $y_0$, $y_1$, $p$) | With a probability of $p$, perform a random shear within the horizontal range of $[x_0, x_1]$ and the vertical range of $[y_0, y_1]$. |
| RandomAffine($t_0$, $t_1$, $x_0$, $y_0$, $t_2$, $t_3$, $x_1$, $x_2$, $y_1$, $y_2$) | Apply a random affine, including rotation between $[t_0, t_1]$ degrees, translation within $[-x_0, x_0]$ horizontally and $[-y_0, y_0]$ vertically, scaling between $[t_2, t_3]$, and shearing horizontally between $[x_1, x_2]$ and vertically between $[y_1, y_2]$. |
| RandomBrightness($t_0$, $t_1$, $p$) | With a probability of $p$, apply a random brightness adjustment within the range of $[t_0, t_1]$, allowing values from -1 to 1, and 0 means no brightness adjustment. |
| RandomSaturation($t_0$, $t_1$, $p$) | With a probability of $p$, apply a random saturation adjustment within the range of $[t_0, t_1]$, allowing values from -1 to 1, and 0 means no saturation adjustment. |
| RandomContrast($t_0$, $t_1$, $p$) | With a probability of $p$, apply a random contrast adjustment within the range of $[t_0, t_1]$, allowing values from -1 to 1, and 0 means no contrast adjustment. |
| RandomSharpness($t_0$, $t_1$, $p$) | With a probability of $p$, apply a random sharpness adjustment within the range of $[t_0, t_1]$, allowing values from -1 to 1, and 0 means no sharpness adjustment. |
| RandomHue($t_0$, $t_1$, $p$) | With a probability of $p$, apply a random hue adjustment within the range of $[t_0, t_1]$, allowing values from -0.5 to 0.5, and 0 means no hue adjustment. |
| RandomSolarize($t_0$, $t_1$, $p$) | With a probability of $p$, generate a threshold within the range of $[t_0, t_1]$, and invert all pixels above the threshold. The threshold value is in the range of [0, 1]. |
| RandomPosterize($t_0$, $t_1$, $p$) | With a probability of $p$, generate a positive integer within the range of $[t_0, t_1]$, and adjust the bit depth of all color channels to that value. |
| RandomEqualize($p$) | With a probability of $p$, equalize the histogram of the image. |
| RandomAutocontrast($p$) | With a probability of $p$, autocontrast the pixels of the image. |
| RandomHorizontalFlip($p$) | With a probability of $p$, horizontally flip the image. |
| RandomErasing($p$, $t$) | Randomly erase a square region of the image, with probability $p$, using an area fraction $t$ between 0 and 1. |

Table 2 presents all the data augmentation techniques utilized in our experiments. Below, we will provide detailed descriptions of the parameters and application specifics for each augmentation technique employed on the MNIST, CIFAR-10/100, and SVHN datasets.

In the experiments conducted on the MNIST dataset, the following four data augmentations were applied in random order:

- RandomRotation($-10$, 10, 1.0)

- RandomTranslation(1, 1, 1.0)

- RandomScaling(0.9, 1.1, 1.0)

- RandomShear($-10$, 10, $-10$, 10, 1.0)

On CIFAR-10/100, we employed an identical data augmentation strategy. Firstly, we applied RandomHorizontalFlip(0.5) to randomly flip images horizontally. Next, we padded the edges of the images by 16 pixels using reflection mode. Subsequently, the following four data augmentations were applied in a random order:

- RandomRotation($-30$, 30, 0.3)
- RandomTranslation(8, 8, 0.3)
- RandomScaling(0.9, 1.1, 0.3)
- RandomShear($-30$, 30, $-30$, 30, 0.3)

To minimize image distortion, we utilized the bicubic interpolation algorithm in this context. After applying the spatial transformations mentioned above, we cropped images to a size of $32 \times 32$ pixels from their centers. Subsequently, the following data augmentations were performed in a random order:

- RandomBrightness($-0.25$, 0.25, 1.0)
- RandomContrast($-0.5$, 0.5, 1.0)
- RandomSharpness($-0.5$, 0.5, 1.0)
- RandomSaturation($-1.0$, 1.0, 1.0)
- RandomSolarize(0.0, 1.0, 0.2)
- RandomPosterize(4, 8, 0.2)
- RandomEqualize(0.2)
- RandomAutocontrast(0.2)

Finally, we employed RandomErasing(1.0, 0.1) to randomly erase $10\%$ of the pixels.

For the SVHN dataset, we padded the edges of the images by 8 pixels using reflection mode. Next, we applied RandomAffine($-10$, 10, 1, 1, 0.9, 1.1, $-10$, 10, $-10$, 10). Subsequently, we cropped images to a size of $32 \times 32$ pixels from their centers. Following the cropping step, the following data augmentations were performed in a random order:

- RandomHue($-0.5$, 0.5, 1.0)
- RandomBrightness($-0.25$, 0.25, 0.5)
- RandomContrast($-0.5$, 0.5, 0.5)
- RandomSharpness($-0.5$, 0.5, 0.5)
- RandomSaturation($-1.0$, 1.0, 0.5)
- RandomSolarize(0.0, 1.0, 0.2)
- RandomPosterize(4, 8, 0.2)
- RandomEqualize(0.2)
- RandomAutocontrast(0.2)

Ultimately, we applied RandomErasing(1.0, 0.2).

The principles followed by the above data augmentation strategies were to generate as much diversity as possible while minimizing any potential damage to the images. However, it is important to note that despite these efforts, these data augmentations may still fall short in accurately simulating the variations that these entities may encounter in the real world.

## B INCONSISTENCIES BETWEEN EXCOST-GENERATED PSEUDO-LABELS AND GROUND TRUTH LABELS

As shown in Figure 2, the $i$-th row in the figure represents pseudo-labels generated by EXCOST for the $i$-th class that are inconsistent with ground truth labels. Upon observation, it becomes evident

Figure 2: Pseudo-labels generated by EXCOST are inconsistent with ground truth labels. The final epoch of training on MNIST with a seed value of 0.

that many of these cases are a result of human annotation errors, and there are also several instances of ambiguous and low-quality samples. Given that similar situations exist in the test dataset and considering that the proposed model achieves an average error rate of only $0.32\%$ on the test set, it is possible that there is limited room for further improvement.

## C  DETAILS OF ABLATION STUDY

Table 3: Error rates on MNIST with 20 labels using CIL when considering exemplar similarity in the labeling algorithm.

| Settings | | | Minimum of All Epochs | Median of Last 20 Epochs |
|---|---|---|---|---|
| $\mathcal{S}_1$ | $\mathcal{S}_3$ | $\mathcal{S}_6$ | 0.327±0.031 | 0.450±0.129 |
| $\mathcal{S}_1$ | $\mathcal{S}_3$ | $\mathcal{S}_7$ | 0.325±0.030 | 0.410±0.054 |
| $\mathcal{S}_2$ | $\mathcal{S}_3$ | $\mathcal{S}_6$ | 0.489±0.083 | 1.214±0.677 |
| $\mathcal{S}_2$ | $\mathcal{S}_3$ | $\mathcal{S}_7$ | 0.505±0.066 | 1.229±0.576 |
| $\mathcal{S}_1$ | $\mathcal{S}_4$ | $\mathcal{S}_6$ | 0.324±0.022 | 0.389±0.057 |
| $\mathcal{S}_1$ | $\mathcal{S}_4$ | $\mathcal{S}_7$ | 0.328±0.022 | 0.409±0.063 |
| $\mathcal{S}_2$ | $\mathcal{S}_4$ | $\mathcal{S}_6$ | 0.426±0.057 | 0.624±0.213 |
| $\mathcal{S}_2$ | $\mathcal{S}_4$ | $\mathcal{S}_7$ | 0.410±0.069 | 0.605±0.179 |
| $\mathcal{S}_1$ | $\mathcal{S}_5$ | $\mathcal{S}_6$ | 0.360±0.085 | 0.498±0.202 |
| $\mathcal{S}_1$ | $\mathcal{S}_5$ | $\mathcal{S}_7$ | 0.368±0.082 | 1.052±1.132 |
| $\mathcal{S}_2$ | $\mathcal{S}_5$ | $\mathcal{S}_6$ | 0.443±0.058 | 0.684±0.160 |
| $\mathcal{S}_2$ | $\mathcal{S}_5$ | $\mathcal{S}_7$ | 0.422±0.082 | 0.678±0.443 |

In Section 4.3, we provided a description of the experimental settings and basic conclusions regarding the ablation study. In this section, we will elaborate further on these details. Table 3 presents the results when using CIL and considering exemplar similarity in the labeling algorithm. Although the differences in average minimum error rates among different experimental settings are not substantial, they do have some impact on result stability. Notably, when both experimental settings $\mathcal{S}1$ and $\mathcal{S}4$ were applied simultaneously, the standard deviation of the minimum error rates was the smallest among all different experimental configurations, at $0.022$. Conversely, when experimental settings $\mathcal{S}2$ and $\mathcal{S}3$ were used concurrently, the median error rates of the last 20 epochs were significantly higher. These patterns are also reflected in Table 4, which excludes considering exemplar similarity in the labeling algorithm. Table 5 and Table 6 present results obtained without using CIL, where it is evident that these error rates are significantly higher compared to the scenarios where CIL is employed.

We present the ablation experiments on exemplar similarity in Table 7, conducted on the CIFAR-10 dataset with 250 labels. The results indicate that, when a loose constraint on exemplar similarity is applied, the error rate is significantly lower compared to the other two conditions. We

Table 4: Error rates on MNIST with 20 labels using CIL when not considering exemplar similarity in the labeling algorithm.

| Settings | | | | Minimum of All Epochs | Median of Last 20 Epochs |
|---|---|---|---|---|---|
| $S_1$ | $S_3$ | $S_6$ | $S_9$ | 0.354±0.046 | 0.545±0.202 |
| $S_1$ | $S_3$ | $S_7$ | $S_9$ | 0.353±0.046 | 0.538±0.206 |
| $S_2$ | $S_3$ | $S_6$ | $S_9$ | 0.498±0.124 | 1.150±0.368 |
| $S_2$ | $S_3$ | $S_7$ | $S_9$ | 0.497±0.111 | 1.131±0.386 |
| $S_1$ | $S_4$ | $S_6$ | $S_9$ | 0.328±0.035 | 0.400±0.057 |
| $S_1$ | $S_4$ | $S_7$ | $S_9$ | 0.334±0.032 | 0.410±0.069 |
| $S_2$ | $S_4$ | $S_6$ | $S_9$ | 0.415±0.040 | 0.789±0.391 |
| $S_2$ | $S_4$ | $S_7$ | $S_9$ | 0.406±0.048 | 0.832±0.463 |
| $S_1$ | $S_5$ | $S_6$ | $S_9$ | 0.332±0.059 | 0.421±0.113 |
| $S_1$ | $S_5$ | $S_7$ | $S_9$ | 0.342±0.053 | 0.459±0.141 |
| $S_2$ | $S_5$ | $S_6$ | $S_9$ | 0.475±0.072 | 1.057±0.606 |
| $S_2$ | $S_5$ | $S_7$ | $S_9$ | 0.490±0.058 | 1.090±0.325 |

Table 5: Error rates on MNIST with 20 labels without using CIL when considering exemplar similarity in the labeling algorithm.

| Settings | | | Minimum of All Epochs | Median of Last 20 Epochs |
|---|---|---|---|---|
| $S_3$ | $S_6$ | $S_8$ | 1.832±0.396 | 1.928±0.398 |
| $S_3$ | $S_7$ | $S_8$ | 1.834±0.397 | 1.935±0.401 |
| $S_4$ | $S_6$ | $S_8$ | 1.599±0.568 | 1.713±0.609 |
| $S_4$ | $S_7$ | $S_8$ | 1.476±0.660 | 1.712±0.897 |
| $S_5$ | $S_6$ | $S_8$ | 1.660±0.419 | 1.733±0.427 |
| $S_5$ | $S_7$ | $S_8$ | 1.415±0.445 | 1.620±0.473 |

Table 6: Error rates on MNIST with 20 labels without using CIL when not considering exemplar similarity in the labeling algorithm.

| Settings | | | | Minimum of All Epochs | Median of Last 20 Epochs |
|---|---|---|---|---|---|
| $S_3$ | $S_6$ | $S_8$ | $S_9$ | 1.837±0.440 | 1.939±0.449 |
| $S_3$ | $S_7$ | $S_8$ | $S_9$ | 1.827±0.434 | 1.932±0.445 |
| $S_4$ | $S_6$ | $S_8$ | $S_9$ | 1.709±0.426 | 1.831±0.449 |
| $S_4$ | $S_7$ | $S_8$ | $S_9$ | 1.694±0.438 | 1.828±0.473 |
| $S_5$ | $S_6$ | $S_8$ | $S_9$ | 1.707±0.430 | 1.789±0.448 |
| $S_5$ | $S_7$ | $S_8$ | $S_9$ | 1.715±0.439 | 1.895±0.508 |

Table 7: Error rates on CIFAR-10 with 250 labels.

| Settings | | | | Minimum of All Epochs | Median of Last 20 Epochs |
|---|---|---|---|---|---|
| $S_1$ | $S_4$ | $S_6$ | | 4.933±0.095 | 5.163±0.147 |
| $S_1$ | $S_4$ | $S_6$ | $S_9$ | 4.967±0.754 | 5.190±0.711 |
| $S_1$ | $S_{10}$ | $S_6$ | | 4.560±0.180 | 4.743±0.156 |

attribute this phenomenon to the substantial dissimilarity among samples in the CIFAR-10 dataset. Since manually designed data augmentations may not yield very effective generalization due to the dataset's inherent diversity, imposing a high degree of similarity between samples and exemplars would overly restrict the model's generalization. Conversely, eliminating the need for similarity leads to a higher risk of confirmation bias. Therefore, in this context, a loose constraint on exem-

plar similarity yields superior results. Nonetheless, we believe that if data augmentations can better mimic transformations in natural scenes, enhancing the model's generalization capabilities, it may generate more abstract category representations. In such a scenario, a relatively tighter constraint on exemplar similarity might yield even better results.

## D REQUIREMENT FOR INVARIANCE IN FEATURE VECTORS

Table 8: Comparison of invariance requirements on class probabilities and feature vectors using experimental settings $\{\mathcal{S}1, \mathcal{S}4, \mathcal{S}6\}$ on the MNIST dataset with 20 labels.

| Loss | Minimum of All Epochs | Median of Last 20 Epochs |
|---|---|---|
| Using CIL | 0.324±0.022 | 0.389±0.057 |
| Requiring Feature Vectors Invariance | 1.558±0.538 | 2.623±1.432 |

Table 9: Comparison of invariance requirements on class probabilities and feature vectors using experimental settings $\{\mathcal{S}1, \mathcal{S}10, \mathcal{S}6\}$ on the CIFAR-10 dataset with 250 labels.

| Loss | Minimum of All Epochs | Median of Last 20 Epochs |
|---|---|---|
| Using CIL | 4.560±0.180 | 4.743±0.156 |
| Requiring Feature Vectors Invariance | 5.280±0.372 | 5.563±0.462 |

In our experiments, we employed the experimental settings $\{\mathcal{S}1, \mathcal{S}4, \mathcal{S}6\}$ on the MNIST dataset with 20 labels (for specific experimental setup settings, please refer to Section 4.3). We compared the results between utilizing CIL (i.e., requiring network outputs' class probabilities to be invariant under different views of the same instance) and shifting the contrast operation to feature vectors, as shown in Table 8. We also conducted experiments on the CIFAR-10 dataset with 250 labels with the experimental settings $\{\mathcal{S}1, \mathcal{S}10, \mathcal{S}6\}$, as presented in Table 9. In both scenarios, it is evident that requiring feature vectors to be invariant results in a significant increase in error rates compared to using CIL.

## E LABELING ERRORS OF PSEUDO-LABEL GENERATION

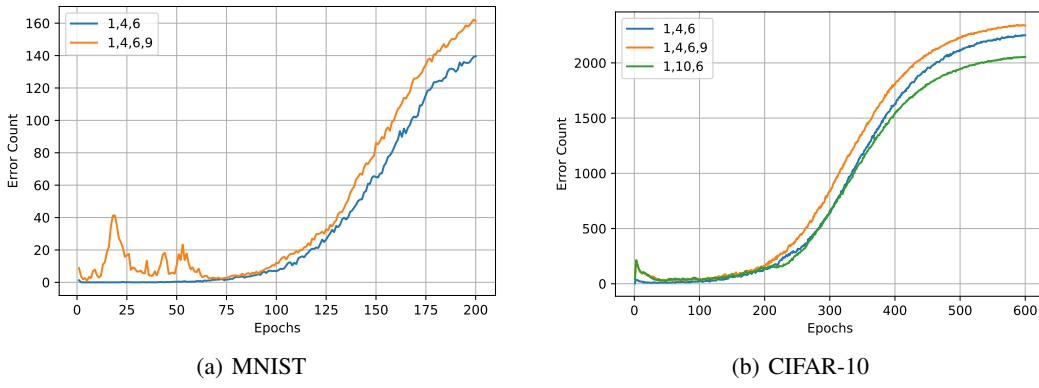

(a) MNIST                    (b) CIFAR-10

Figure 3: Comparison of labeling errors under different experimental settings for the labeling algorithm.

For the sake of clarity, we will keep the experimental settings description as presented in Section 4.3. In Figure 3(a), we depict the average labeling errors using ten different random seeds under two sets of experimental conditions: $\{\mathcal{S}1, \mathcal{S}4, \mathcal{S}6\}$, and $\{\mathcal{S}1, \mathcal{S}4, \mathcal{S}6, \mathcal{S}9\}$. It's noteworthy that fewer

labeling errors occur when $\mathcal{S}9$ is not utilized. Specifically, during the early stages of the training process, a significant number of labeling errors are generated when we do not consider exemplar similarity in the labeling algorithm. This problem is nearly eliminated when we take exemplar similarity into account.

We observed similar trends in the CIFAR-10 dataset, as depicted in Figure 3(b). In this context, we also present the results using experimental settings $\{\mathcal{S}1, \mathcal{S}10, \mathcal{S}6\}$. The primary distinction between $\mathcal{S}10$ and $\mathcal{S}4$ lies in the relaxed constraint on exemplar similarity. Notably, owing to the initially looser constraint on exemplar similarity within $\{\mathcal{S}1, \mathcal{S}10, \mathcal{S}6\}$, they exhibit errors similar to those seen in $\{\mathcal{S}1, \mathcal{S}4, \mathcal{S}6, \mathcal{S}9\}$ during the early stages of training. However, as training progresses, $\{\mathcal{S}1, \mathcal{S}10, \mathcal{S}6\}$ consistently achieve the lowest labeling errors among these three settings.

## F  DISCUSSION

EXCOST refrains from employing specific forms of data augmentation, such as the concurrent utilization of strong and weak augmentations required by FixMatch, or the requirement of MixUp for MixMatch. These methodologies, while influential, introduce certain constraints and intuitively unnatural conditions. Notably, when considering the MNIST dataset, elucidating the distinction between weak and strong augmentations becomes challenging within the FixMatch framework. Similarly, the utilization of MixUp introduces the possibility of altering the inherent meaning of original images, as evident in instances where two zeros could amalgamate into the representation of an eight.

In contrast, EXCOST operates by tailoring data augmentation strategies according to the specifics of the dataset, an approach that imbues models with informative priors. A prime example is the consideration of MNIST and SVHN, which are less suited for horizontal flips, in contrast to the appropriateness of such augmentations for CIFAR-10/100. The justification for these augmentations stems from the rationality inherent in the real-world context. Crucially, effective data augmentation preserves the intrinsic meaning of samples, exemplifying invariance. Remarkably, humans have consistently been learning these invariant features from birth. Human perception processes continuously present a sequence of coherent visual frames, and the variations in each frame related to a particular instance can be viewed as a form of natural data augmentation. This process of augmentation bestows models with richer information, encapsulating the operational regularities of the real world—an essential element in fostering the model's reasoning abilities.

Currently, the model utilizes static image datasets for semi-supervised image classification. However, real-world information is often not confined to static images but consists of continuous video frames. Consequently, manually designed data augmentation strategies may not fully capture the dynamic variations between images. To address this issue, we can consider training the model using video data instead of solely relying on static images. By using adjacent or closely spaced video frames, the model can learn the patterns of change inherent in continuous animations. For instance, in a video of a person walking, not only does the position change but there are also variations in the movement of the legs. By training the model to comprehend these continuous changes, we can enable the model to better capture the invariances present in the real world without the need for manually crafting specific data augmentation strategies.

