# OpenReview forum: "EXCOST: Semi-Supervised Classification with Exemplar-Contrastive Self-Training"
_ICLR.cc/2024/Conference — Submitted to ICLR 2024_

### Official Review · Reviewer_nk72 · 2023-10-29

**Soundness:** 2 fair
**Presentation:** 2 fair
**Contribution:** 2 fair
**Rating:** 5
**Confidence:** 4

**Summary:**

This paper introduces a novel semi-supervised learning (SSL) algorithm called Exemplar-Contrastive Self-Training (EXCOST). The primary goal of this algorithm is to enhance the reliability of pseudo-labels by considering both high confidence and exemplar similarity. Additionally, the paper presents a unique regularization term known as Category-Invariant Loss (CIL), which aims to encourage consistent class probabilities for the same sample under various perturbations.

**Strengths:**

1. The proposed approach is independent of prevailing weak and strong data augmentation strategies and does not rely on the use of exponential moving averages.
2. The effectiveness of the proposed EXCOST is demonstrated through comprehensive evaluations in semi-supervised image classification tasks.

**Weaknesses:**

1. Integrating the concept of contrastive learning into semi-supervised learning is not a novel approach, and several similar studies have been conducted previously [1,2,3,4]. The authors should analyze and experimentally compare their method with these previous approaches.

2. As mentioned in the previous question, the Related Work in this paper is not comprehensive, lacks a summary and analysis of prior work, and fails to highlight the contributions of this paper.

3. The experimental section lacks an introduction to baseline methods, and many experimental results are missing from Table 1. It would be preferable for the authors to mark the missing results with horizontal lines.

4. The author's expression is inconsistent. In the first paragraph of the Introduction, the author states that the exemplars theory emphasizes the uniqueness of different samples within the same category. However, in Section 3.2, the author claims that the exemplars theory suggests that samples of the same class should be similar, and pseudo-labels are calculated based on the similarity between samples and exemplars. It is unclear how the uniqueness of samples is reconciled with this.

5. The nomenclature in this paper should be more standardized. For instance, do ${\Phi _c}$ and ${\phi _c}$ represent the same thing?

6. The article is difficult to understand, and the authors should restate the motivation of this paper and clarify the relationship between motivation and the proposed method. For example, how are prototype theory and exemplar theory reflected in the methodology, and what role does the typicality gradient play in this paper?

[1 ]Li J, Xiong C, Hoi S C H. Comatch: Semi-supervised learning with contrastive graph regularization[C]//Proceedings of the IEEE/CVF International Conference on Computer Vision. 2021: 9475-9484.

[2] Zheng M, You S, Huang L, et al. Simmatch: Semi-supervised learning with similarity matching[C]//Proceedings of the IEEE/CVF Conference on Computer Vision and Pattern Recognition. 2022: 14471-14481.

[3] Zhang Y, Zhang X, Li J, et al. Semi-supervised contrastive learning with similarity co-calibration[J]. IEEE Transactions on Multimedia, 2022.

[4] Zheng M, You S, Huang L, et al. SimMatchV2: Semi-Supervised Learning with Graph Consistency[C]//Proceedings of the IEEE/CVF International Conference on Computer Vision. 2023: 16432-16442.

**Questions:**

Please refer to Weaknesses

---

> ### Author Response · Authors · 2023-11-17
> **Response to Reviewer nk72 (1/3)**
>
> Thank you for your thorough and insightful review of our manuscript. We appreciate the time and effort you have dedicated to providing valuable feedback on our work, and here is our response:
>
>
> > Integrating the concept of contrastive learning into semi-supervised learning is not a novel approach, and several similar studies have been conducted previously [1,2,3,4]. The authors should analyze and experimentally compare their method with these previous approaches.
>
> Thank you very much for providing the references. We will include an introduction to [1,2,3,4] in the related work section of the next version.
>
>
> > As mentioned in the previous question, the Related Work in this paper is not comprehensive, lacks a summary and analysis of prior work, and fails to highlight the contributions of this paper.
>
> Thank you once again for pointing out our issue. We will include [1,2,3,4] in our related work section to provide a more comprehensive summary of previous work.
>
>
> > The experimental section lacks an introduction to baseline methods, and many experimental results are missing from Table 1. It would be preferable for the authors to mark the missing results with horizontal lines.
>
> We used the raw data from the original papers of all baseline models in Table 1. Since this data was missing in the original papers, we left those parts blank to ensure the accuracy of the information. We appreciate your valuable suggestion. We will take it into account and use horizontal lines to fill in the blank parts.
>
>
> > The author's expression is inconsistent. In the first paragraph of the Introduction, the author states that the exemplars theory emphasizes the uniqueness of different samples within the same category. However, in Section 3.2, the author claims that the exemplars theory suggests that samples of the same class should be similar, and pseudo-labels are calculated based on the similarity between samples and exemplars. It is unclear how the uniqueness of samples is reconciled with this.
>
> We sincerely apologize for the confusion caused by our lack of clarity in expressing ourselves. Allow us to address your concerns: It is precisely because different samples possess distinctiveness (features beyond those emphasized by prototype) that classifying based on the similarity of different samples is reliable. Let us provide an example using the CIFAR-10 dataset to help illustrate this concept. The CIFAR-10 dataset contains various images of cars, including some samples that are consecutive frames of the same car at different moments in motion. This particular car possesses features that distinguish it from other cars, making it distinctive. In fact, these images may not even be from the same car but rather from the same car model, thereby sharing similarities. If we identify one of these samples as belonging to the "car" class and calculate similarity based on their features, we can be highly confident that the other samples also belong to the "car" class. In other words, if two unique samples happen to be similar, there is a high likelihood that they belong to the same category. Within the same category, similarity exists in a continuous space, forming the basis of a typicality gradient where samples closer to the prototypes are more typical.
>
>
> > The nomenclature in this paper should be more standardized. For instance, do $\Phi_C$ and $\phi_C$ represent the same thing?
>
> Thank you very much for your correction. We will investigate the issue of symbol standardization. $\Phi_C$ and $\phi_C$ represent different entities, where $\Phi_C$ is a function that generates a real value $\phi_C$ as its dependent variable based on the progress of training. Similar cases include $\Psi$ and $\psi$, as well as $\Phi_S$ and $\phi_S$. We will provide further clarification on their relationships in Sections 3.1 and 3.2.

---

> ### Author Response · Authors · 2023-11-17
> **Response to Reviewer nk72 (2/3)**
>
> > The article is difficult to understand, and the authors should restate the motivation of this paper and clarify the relationship between motivation and the proposed method. For example, how are prototype theory and exemplar theory reflected in the methodology, and what role does the typicality gradient play in this paper?
>
> Thank you very much for your suggestions, and I will make an effort to improve the clarity of this paper. Below, we will address the questions you raised:
>
> The motivation of this paper is to simulate the human classification process at the psychological level. Therefore, we utilize the concepts of prototypes and exemplars, combining their strengths to obtain more reliable pseudo-labels. In the labeling algorithm of EXCOST, we always select the sample that is most in line with the prototype (considering the attribute-frequency theory [5,6], i.e., the sample with the highest confidence score from the network output) and has the highest similarity to the exemplars to assign pseudo-labels. The number of selected samples is controlled by the functions $\Phi_C$ and $\Phi_S$, which are monotonically increasing functions with respect to training progress (ranging from 0 to 1). Consequently, the model can utilize more pseudo-labels as the training progresses. This process is analogous to human learning, where the ability to make accurate classification decisions increases as one becomes more familiar with a certain category, while it is limited when facing less familiar instances.
>
> The concepts of prototypes and exemplars are reflected in the labeling algorithm. For the prototype theory, we adopt the attribute-frequency theory, which we believe is equivalent to the confidence generated by the neural network. This is because neural networks optimize through the backpropagation algorithm, and the connections between features that frequently co-occur with a particular class label are strengthened. Therefore, we can infer that the more frequently a feature co-occurs with a class label, the higher the corresponding class probability will be. For the exemplar theory, we treat samples with true labels as exemplars and determine whether they belong to the same category by comparing the similarity between samples and exemplars.
>
> The typicality gradient is mainly reflected in Category-Invariant Loss (CIL). Unlike the FixMatch algorithm [7], which uses hard pseudo-labels, CIL uses class probabilities generated from the original samples instead of using one-hot vectors. Consider the following scenario: if there is no label information in the training set, the model cannot generate class probabilities for samples through learning. Therefore, it can be inferred that the class probabilities of unlabeled samples are gradually learned through the guidance of labeled samples. According to the attribute-frequency theory, all supervised learning on labeled samples will cause the most confident samples to be closest to the prototypes. Thus, the class probability vector can be regarded as a representation of the typicality of a sample's category. We hope that the inputs with data augmentation can be close to this representation, rather than approaching the one-hot vector (losing typicality). Therefore, we believe that CIL contributes to the preservation of typicality gradients.
>
> For samples that are ambiguous in classification, when there is not enough information for the model to make a decision, we hope these samples can stay as close as possible to their original positions instead of actively moving towards the center of any category. The benefit of doing so is that when the model obtains more information about these ambiguous samples in the future, it can learn faster and avoid making excessive efforts to correct mistakes. Moreover, preserving the typicality gradient also provides basic information for distinguishing intra-class members. For example, penguins belong to the bird category, but we can all recognize their differences from other birds. Of course, in reality, our algorithm still falls short of achieving this goal, mainly because we artificially set functions ($\Phi_C$ and $\Phi_S$) to control the labeling rate and force the model to make choices at specified moments. However, in future work, combining with active learning or reinforcement learning may achieve a more human-like psychological behavior.
>
>
> We have taken careful note of your comments and are committed to making the necessary revisions. Should you have any further inquiries or require additional clarification, we welcome the opportunity for continued discussion, and look forward to addressing your concerns in our upcoming revisions. Thank you once again for your constructive feedback.

---

> ### Author Response · Authors · 2023-11-17
> **Response to Reviewer nk72 (3/3)**
>
> ## Reference
>
> [1] Junnan Li, Caiming Xiong, and Steven C.H. Hoi. CoMatch: Semi-supervised learning with contrastive graph regularization. In Proceedings of the IEEE/CVF International Conference on Computer Vision (ICCV), pp. 9475–9484, 2021.
>
> [2] Mingkai Zheng, Shan You, Lang Huang, Fei Wang, Chen Qian, and Chang Xu. SimMatch: Semi-supervised learning with similarity matching. In Proceedings of the IEEE/CVF Conference on Computer Vision and Pattern Recognition (CVPR), pp. 14471–14481, 2022.
>
> [3] Yuhang Zhang, Xiaopeng Zhang, Jie Li, Robert C. Qiu, Haohang Xu, and Qi Tian. Semi-supervised contrastive learning with similarity co-calibration. IEEE Transactions on Multimedia, 25:1749–1759, 2023.
>
> [4] Mingkai Zheng, Shan You, Lang Huang, Chen Luo, Fei Wang, Chen Qian, and Chang Xu. SimMatchV2: Semi-supervised learning with graph consistency. In Proceedings of the IEEE/CVF International Conference on Computer Vision (ICCV), pp. 16432–16442, 2023.
>
> [5] Paul G. Neumann. An attribute frequency model for the abstraction of prototypes. Memory & Cognition, 2(2):241–248, 1974.
>
> [6] Robert L. Solso and Judith E. McCarthy. Prototype formation: Central tendency model vs. attribute-frequency model. Bulletin of the Psychonomic Society, 17(1):10–11, 1981.
>
> [7] Kihyuk Sohn, David Berthelot, Nicholas Carlini, Zizhao Zhang, Han Zhang, Colin A Raffel, Ekin Dogus Cubuk, Alexey Kurakin, and Chun-Liang Li. FixMatch: Simplifying semi-supervised learning with consistency and confidence. In Advances in Neural Information Processing Systems, volume 33, pp. 596–608. Curran Associates, Inc., 2020.

---

> > ### Comment · Reviewer_nk72 · 2023-11-22
> > **Response to authors**
> >
> > The authors have addressed some of my concerns. I still feel confused to understand the relationship between motivation and the proposed method.  All some mentioned experimental results are not provided. So, I keep my rating.

---

### Official Review · Reviewer_SpjF · 2023-10-30

**Soundness:** 3 good
**Presentation:** 2 fair
**Contribution:** 2 fair
**Rating:** 3
**Confidence:** 4

**Summary:**

Semi-supervised learning (SSL) is a hot topic in machine learning. To generate high-quality pseudo-labels for unlabeled data, this paper proposes an Exemplar-Contrastive Self-Training (EXCOST) method along with a category-invariant loss for the model’s training.

**Strengths:**

+ An exemplar-contrastive self-training method for SSL is proposed to achieve trust-worthy pseudo-labeling.
+ A category-invariant loss is designed to encourage models to produce the similar probability distribution for different-view samples.
+ Experiments on several benchmarks have demonstrated the effectiveness of the proposed method.

**Weaknesses:**

- The motivation is not clear. Why combine prototype and exemplar information to improve the quality of pseudo-labels? The first paragraph does not explain well the role of this combination in semi-supervised learning.
- The novelty of this manuscript seems to be limited. The authors should discuss the relation between existing methods and the proposed method in details.
- The organization and the writing of this manuscript should be largely improved for a clear understanding.
- The authors claim that as training progresses, more pseudo-labeled samples are incorporated into the learning process. However, there are no results to prove the point.
- The ablation study is confusing. Moreover, most of the results in Table 1 remain blank.
- Some experimental details are missing, e.g., the threshold in Eq. (1).

**Questions:**

Please refer to the Weaknesses.

---

> ### Author Response · Authors · 2023-11-17
> **Response to Reviewer SpjF (1/3)**
>
> Thank you for your detailed and constructive feedback on our manuscript. We appreciate the time and effort you invested in reviewing our work. Your insights are invaluable in helping us enhance the quality of our contribution. Our responses to your comments are given below:
>
>
> > The motivation is not clear. Why combine prototype and exemplar information to improve the quality of pseudo-labels? The first paragraph does not explain well the role of this combination in semi-supervised learning.
>
> Thank you for your reminder. We may not have adequately explained why we took this approach. Our primary motivation is to guide semi-supervised learning from a psychological perspective by simulating the way humans categorize things. In cognitive psychology, prototypes and exemplars are used to guide human categorization, each having its own advantages and disadvantages. Specifically:
>
> - Prototypes can be considered as the average of features within a category, and the individuality of different samples is lost through this averaging operation. Therefore, prototypes retain the commonalities of samples within that category, making samples with these commonalities more easily discernible as belonging to that category.
> - Exemplars are each entity we encounter, and these entities carry comprehensive information, thereby retaining the individuality of each entity. These individual characteristics may not be common features across the entire category, but they may play a role in classification, especially when categorizing atypical samples.
> - Assessing atypical members of a category through prototypes can be challenging. People may question whether "penguins belong to the bird category" because penguins are atypical members of birds and do not possess all the emphasized features of birds, such as the ability to fly.
> - Exemplars retain too many detailed features, making them less generalizable than prototypes.
>
> We can illustrate this with a practical example using the MNIST dataset: The digit 7 has some distinctive variations in its writing, with a horizontal line often present in the middle position. If we average the features of all samples belonging to the class of digit 7, this specific horizontal line may be overlooked as noise due to its infrequent occurrence. Depending on whether this distinctive variation is randomly selected as an exemplar (labeled sample), we will discuss the following two scenarios:
>
> 1. If a sample with this specific variation is randomly chosen as an exemplar, the features of this exemplar will be quite similar to the features of all other samples with the same distinctive variation. This ensures that digits with this special writing style of 7 will not be misclassified into the wrong category.
> 2. If a 7 with this specific writing style is not selected as an exemplar, it still ensures that these samples will not be prematurely classified into incorrect categories, as their features are dissimilar to any other category's exemplars. As the model matures through learning and with the increase in labeled data, there is a higher probability that the model will correctly classify these samples when forced to do so. This is akin to the human tendency to follow the principle of starting with easier tasks before tackling more challenging ones. In fact, when this special writing style of 7 is not included as an exemplar, there is not sufficient reason to assert that it belongs to the category of 7; the attribution of this writing style to the category of 7 is, in fact, a human-defined convention. Therefore, even in teaching humans, it would be appropriate to showcase this specific style of writing 7 to illustrate this point.
>
> In summary, we aim to integrate prototypes and exemplars to jointly constrain the process of generating pseudo-labels, seeking to obtain higher-quality pseudo-labels and thereby reduce the emergence of confirmation bias. Thank you once again for your guidance, and we will revise the first paragraph in the next version to enhance the clarity of our motivation.

---

> ### Author Response · Authors · 2023-11-17
> **Response to Reviewer SpjF (2/3)**
>
> > The novelty of this manuscript seems to be limited. The authors should discuss the relation between existing methods and the proposed method in details.
>
> Thank you very much for your valuable comments on our paper. Regarding the issue of the manuscript's innovativeness that you raised, we fully understand your concerns. We would like to emphasize here that, while you may perceive its innovativeness to be limited, we firmly believe that our approach exhibits unique innovation in specific aspects. The inspiration for our proposed method is drawn from relevant theories of category representation in cognitive psychology, with the incorporation of machine learning techniques such as self-training, consistency regularization, and contrastive learning. To illustrate the connections and distinctions between our model and the following models, we provide examples:
>
> - Both UDA [1] and EXCOST incorporate the idea of consistency regularization. UDA calculates the cross-entropy between the predicted distributions of augmented samples and original samples as a form of consistency regularization loss. On the other hand, EXCOST achieves consistency by calculating the Category-Invariant Loss (CIL).
> - Both MPL [2] and EXCOST draw inspiration from the idea of self-training. The key idea involves the teacher network continuously learning from the feedback of the student network to generate pseudo-labels that best assist the learning of the student network. In the case of EXCOST, reliable pseudo-labels are generated by combining constraints between prototypes and exemplars.
> - FixMatch [3] and EXCOST both incorporate the ideas of self-training and consistency regularization. In the training process, FixMatch generates a prediction for each unlabeled sample through "weak augmentation." If the class probability in the prediction exceeds a threshold, the prediction is transformed into a "hard" pseudo-label. On the other hand, EXCOST generates pseudo-labels during the labeling phase by referencing confidence-based ranking (prototype) and exemplar-similarity-based ranking (exemplar). FixMatch requires the predictions of "strongly augmented" and "weakly augmented" for the same sample to be consistent, utilizing "hard" pseudo-labels and cross-entropy loss. EXCOST employs Category-Invariant Loss (CIL) to ensure consistency and emphasizes retaining typical gradients of classes.
>
> Our approach not only aims to achieve better results in semi-supervised image classification tasks but also places a strong emphasis on simulating human behavior at a psychological level. Therefore, in our subsequent work, we will continuously improve the model to make it more similar to humans. The ultimate vision is to realize a human-like category representation system, providing abstract category concepts for artificial general intelligence.
>
> However, to achieve this goal, we first need to establish a perceptual environment for the model that is similar to the environment encountered by humans, including the form of data and the way information is interacted with. This cannot be limited to semi-supervised image classification tasks because, in the visual aspect, humans encounter continuous scenes rather than static images. In terms of interaction, the task of semi-supervised image classification prescribes a passive way of information interaction, learning through limited information. Humans, on the other hand, usually ask questions when encountering unfamiliar things, and errors in understanding can be corrected by the environment.
>
> Since the data augmentation in EXCOST does not restricted to a specific form (which sets it apart from MixMatch [4] and FixMatch), it is compatible with any form of data augmentation. We consider consecutive frames in a video constitute a "natural" form of data augmentation. The motion changes in these frames contain a lot of important information, forming the basis for the model to understand the real world. Our model aims to construct a category representation system at the psychological level, making it more interpretable.
>
>
> > The organization and the writing of this manuscript should be largely improved for a clear understanding.
>
> Thank you for pointing out the issues in our paper. We will work on improving the clarity of the manuscript. We have structured our paper following the format of previous ICLR conference papers, and we will consider making adjustments as needed.

---

> ### Author Response · Authors · 2023-11-17
> **Response to Reviewer SpjF (3/3)**
>
> > The authors claim that as training progresses, more pseudo-labeled samples are incorporated into the learning process. However, there are no results to prove the point.
>
> Since both scheduling functions regarding the labeling rate (i.e., $\Phi_C$ and $\Phi_S$) are monotonically increasing, the labeling rate will gradually rise as the training progresses. This ensures that more pseudo-labels are incorporated into the learning process. Due to the relative obviousness of this conclusion and constraints on space, we have opted not to provide a formal proof.
>
>
> > The ablation study is confusing. Moreover, most of the results in Table 1 remain blank.
>
> We conducted rigorous ablation experiments on the MNIST dataset, thoroughly exploring various combinations of hyperparameters under different conditions to obtain comprehensive experimental results.
>
> The results in Table 1 are sourced from the original data of baseline models' papers. Since the original data did not include this specific information, there are blank entries. We did not alter any original data to ensure its accuracy. For instance, we did not supplement results for the baseline models with blanks for MNIST, primarily because we couldn't ascertain the optimal hyperparameter settings for these models on this dataset. For example, for FixMatch and its improved models, determining the "weak augmentation" and "strong augmentation" seems less straightforward.
>
>
> > Some experimental details are missing, e.g., the threshold in Eq. (1).
>
> Thank you very much for pointing out our oversight. The symbol $\psi$ is generated by the function $\Psi$, and similarly, $\phi_C$ and $\phi_S$ are produced by the functions $\Phi_C$ and $\Phi_S$, respectively. We will supplement explanations in Sections 3.1 and 3.2 to clarify the relationships among them.
>
>
> We hope that the revisions we have made address the concerns raised, and we look forward to your reevaluation of our manuscript. If you have any further questions or require additional clarification, please feel free to reach out. Your expertise is immensely valuable to us, and we are committed to improving our work based on your valuable feedback.
>
>
> ## Reference
>
> [1] Qizhe Xie, Zihang Dai, Eduard Hovy, Thang Luong, and Quoc Le. Unsupervised data augmentation for consistency training. In Advances in Neural Information Processing Systems, volume 33, pp. 6256–6268. Curran Associates, Inc., 2020a.
>
> [2] Hieu Pham, Zihang Dai, Qizhe Xie, and Quoc V. Le. Meta pseudo labels. In Proceedings of the IEEE/CVF Conference on Computer Vision and Pattern Recognition (CVPR), pp. 11557–11568, 2021.
>
> [3] Kihyuk Sohn, David Berthelot, Nicholas Carlini, Zizhao Zhang, Han Zhang, Colin A Raffel, Ekin Dogus Cubuk, Alexey Kurakin, and Chun-Liang Li. FixMatch: Simplifying semi-supervised learning with consistency and confidence. In Advances in Neural Information Processing Systems, volume 33, pp. 596–608. Curran Associates, Inc., 2020.
>
> [4] David Berthelot, Nicholas Carlini, Ian Goodfellow, Nicolas Papernot, Avital Oliver, and Colin A Raffel. MixMatch: A holistic approach to semi-supervised learning. In Advances in Neural Information Processing Systems, volume 32. Curran Associates, Inc., 2019.

---

### Official Review · Reviewer_N7m5 · 2023-11-03

**Soundness:** 3 good
**Presentation:** 2 fair
**Contribution:** 3 good
**Rating:** 3
**Confidence:** 3

**Summary:**

In this paper, the authors introduce a semi-supervised learning (SSL) algorithm called Exemplar-Contrastive Self-Training (EXCOST) with the aim of enhancing the quality of pseudo-labels for unlabeled samples. Additionally, they propose a Category-Invariant Loss, which encourages the model to produce consistent class probabilities for the same sample under different perturbations.

**Strengths:**

The idea of Exemplar-Contrastive Self-Training is interesting to me.  It generates more reliable pseudo-labels by combining measures of confidence and exemplar similarity.

**Weaknesses:**

1.	Effectiveness: While the concept is interesting, the paper falls short in terms of performance when compared to SOTA methods. Notably, the proposed method exhibits considerably higher error rates on CIFAR-100 in comparison to other techniques.
2.	Writing Quality: The clarity of the paper's presentation could be improved. For instance, in Table 1, there's a mix of reporting styles, with some methods providing the median error rate of the last 20 epochs, while others report the minimum error rate across all epochs. This inconsistency makes it challenging to make direct comparisons.
3.	Experiment Setup and Results: To enhance the paper's organization, it's advisable to present the primary results in the main paper, while relocating additional details, like the results of the ablation study, to the appendix. This will streamline the main paper and maintain a focused narrative.

**Questions:**

Please see Weaknesses

---

> ### Author Response · Authors · 2023-11-17
> **Response to Reviewer N7m5 (1/2)**
>
> Thank you for your thoughtful and constructive feedback on our manuscript. We appreciate the time and effort you dedicated to reviewing our work. Your feedback has provided valuable perspectives that will undoubtedly contribute to the enhancement of our paper. We hope the following explanations will answer your questions.
>
>
> > Effectiveness: While the concept is interesting, the paper falls short in terms of performance when compared to SOTA methods. Notably, the proposed method exhibits considerably higher error rates on CIFAR-100 in comparison to other techniques.
>
> We appreciate your interest in the method proposed in this paper. From Table 1, it can be observed that except for the CIFAR-100 dataset, EXCOST achieves state-of-the-art (SOTA) or comparable performance levels on tasks such as MNIST, CIFAR-10, and SVHN (notably, these results were obtained using the Adam [1] optimization algorithm). However, it is important to note that our experimental conditions were quite limited, with only 4 RTX 3090 GPUs available. Therefore, it was challenging to identify the optimal hyperparameters, especially those related to the optimizer and data augmentation, which can have a significant impact on the results. Since our approach does not directly build upon existing mature models, we had to explore the hyperparameters from scratch. This is particularly challenging when computational resources are limited. However, the scarcity of hardware resources is a common situation faced by many researchers, which greatly reduces the possibility of achieving better experimental results. For the CIFAR-100 dataset, it took approximately 76 hours to validate a set of experimental configurations using one RTX 3090 GPU. Due to resource limitations, we could only adjust the parameters of data augmentation based on the concept of not severely distorting the images, which restricted our ability to discover better results on CIFAR-100.
>
> Additionally, it is worth noting that, following common experimental practices, we provide an equal number of labeled samples for all categories. Specifically, both the training and testing datasets of CIFAR-100 are uniformly distributed across all classes. Therefore, techniques such as Distribution Alignment (DA) from ReMixMatch [2] or self-adaptive fairness (SAF) from FreeMatch [3] yield better results on CIFAR-100. However, their effectiveness on datasets with non-uniform class distributions remains uncertain. For we have abandoned this assumption, we cannot leverage these methods. In Section 4.1 of the FixMatch [4] paper, the third paragraph illustrates the benefits of DA on CIFAR-100 results.
>
>
> > Writing Quality: The clarity of the paper's presentation could be improved. For instance, in Table 1, there's a mix of reporting styles, with some methods providing the median error rate of the last 20 epochs, while others report the minimum error rate across all epochs. This inconsistency makes it challenging to make direct comparisons.
>
> We apologize for not clearly expressing ourselves and causing you to have a bad reading experience. We will make every effort to improve the clarity of our paper. It should be noted that in Table 1, we did not reproduce the baseline model but directly quoted the results from the original paper. This decision was based on two main considerations:
>
> 1. We did not have sufficient computing resources to reproduce all the models.
> 2. The authors of the original paper have the deepest understanding of the models they designed, and we trust the results they have achieved. Additionally, the evaluation methods in these papers vary (i.e., some papers report the minimum error rate of all epochs, while others report the median error rate of the last 20 epochs, and some papers did not provide relevant explanations about the error rate). Therefore, we believe it is necessary to make this distinction clear.

---

> ### Author Response · Authors · 2023-11-17
> **Response to Reviewer N7m5 (2/2)**
>
> > Experiment Setup and Results: To enhance the paper's organization, it's advisable to present the primary results in the main paper, while relocating additional details, like the results of the ablation study, to the appendix. This will streamline the main paper and maintain a focused narrative.
>
> Thank you for your excellent suggestions. We have taken your advice on board, and in the next version of the paper, the ablation study will be moved to the appendix.
>
>
> Once again, we express our gratitude for your meticulous review and constructive feedback. Your insights have been instrumental in shaping the future direction of our work. If you have any further questions or if there are additional aspects of the paper that you would like us to clarify, please do not hesitate to reach out. We welcome the opportunity to engage in further discussion.
>
>
> ## Reference
>
> [1] Diederik P. Kingma and Jimmy Ba. Adam: A method for stochastic optimization, 2017.
>
> [2] David Berthelot, Nicholas Carlini, Ekin D. Cubuk, Alex Kurakin, Kihyuk Sohn, Han Zhang, and Colin Raffel. ReMixMatch: Semi-supervised learning with distribution matching and augmentation anchoring. In International Conference on Learning Representations (ICLR), 2020.
>
> [3] Yidong Wang, Hao Chen, Qiang Heng, Wenxin Hou, Yue Fan, Zhen Wu, Jindong Wang, Marios Savvides, Takahiro Shinozaki, Bhiksha Raj, Bernt Schiele, and Xing Xie. FreeMatch: Self-adaptive thresholding for semi-supervised learning. In The Eleventh International Conference on Learning Representations, 2023.
>
> [4] Kihyuk Sohn, David Berthelot, Nicholas Carlini, Zizhao Zhang, Han Zhang, Colin A Raffel, Ekin Dogus Cubuk, Alexey Kurakin, and Chun-Liang Li. FixMatch: Simplifying semi-supervised learning with consistency and confidence. In Advances in Neural Information Processing Systems, volume 33, pp. 596–608. Curran Associates, Inc., 2020.

---

### Official Review · Reviewer_GnDx · 2023-11-03

**Soundness:** 2 fair
**Presentation:** 2 fair
**Contribution:** 2 fair
**Rating:** 5
**Confidence:** 3

**Summary:**

This paper introduces a semi-supervised learning (SSL) algorithm called Exemplar-Contrastive Self-Training (EXCOST).  EXCOST determines pseudo-labels for unlabeled data with high confidence and exemplar similarity for self-training. The paper also presents a novel regularization term known as Category-Invariant Loss (CIL) to enhance the consistency of class probabilities across different representations of the same sample under various perturbations. The paper achieves state-of-the-art results on semi-supervised image classification tasks across various benchmark datasets, such as MNIST, SVHN, and CIFAR-10.

**Strengths:**

1. The paper is easy to follow and well-structured.
2. The proposed components; Category-Invariant Loss and exemplar-contrastive self-training is novel to some extent.
3. The experiments, including the appendix, are thorough and well-designed, providing comprehensive results across all settings.
4. The listed performance demonstrates the effectiveness of the proposed method, significantly improving performance compared to the previous one.

**Weaknesses:**

1. One potential limitation of the Category-Invariant Loss (CIL) is its sensitivity to the choice of the threshold parameter. The text acknowledges that the threshold is introduced to manage the influence of irrelevant or outlier samples. However, determining the appropriate threshold value may not be straightforward and can significantly impact the loss function.

2. The exemplar-contrastive self-training algorithm involves several hyper-parameters, including thresholds, margin values, and labeling rates. The sensitivity of these hyper-parameters could present challenges during practical implementation, as selecting the right values may require thorough experimentation and tuning.

3. While the paper mentions that the computational burden is reduced by deferring the computation of exemplar feature vectors until the labeling phase, it lacks a comprehensive analysis of the algorithm's computational efficiency. The practical implications and potential computational overhead of the entire algorithm should be subject to in-depth investigation and evaluation.

**Questions:**

Please address my concerns

---

> ### Author Response · Authors · 2023-11-17
> **Response to Reviewer GnDx (1/2)**
>
> Thank you for your thoughtful review of our paper. We appreciate the time and effort you dedicated to providing valuable feedback on our work. We respond to the concerns below:
>
>
> > One potential limitation of the Category-Invariant Loss (CIL) is its sensitivity to the choice of the threshold parameter. The text acknowledges that the threshold is introduced to manage the influence of irrelevant or outlier samples. However, determining the appropriate threshold value may not be straightforward and can significantly impact the loss function.
>
> Thank you for raising this question. In the ablation experiment (Table 3), we were able to compare the cases where the threshold gradually increased from $\frac{1}{10}$ to approximately 0.9 (denoted as $\mathcal{S}_1$) and where the threshold was constantly set at 0 (denoted as $\mathcal{S}_2$, which effectively corresponds to the range of $[0, \frac{1}{10}]$ for a 10-class problem) on the MNIST dataset. In terms of the minimum error rate among all epochs, the average error rate for $\mathcal{S}_1$ was $0.339$, and for $\mathcal{S}_2$, it was $0.449$. In terms of the median error rate of the last 20 epochs, the average error rate for $\mathcal{S}_1$ was $0.535$, while for $\mathcal{S}_2$, it was $0.839$. From this, it can be seen that the threshold does not significantly affect the model's learning but has a slight impact on the stability in the later stages of training.
>
>
> > The exemplar-contrastive self-training algorithm involves several hyper-parameters, including thresholds, margin values, and labeling rates. The sensitivity of these hyper-parameters could present challenges during practical implementation, as selecting the right values may require thorough experimentation and tuning.
>
> This is a great question, and it's a challenge faced by many deep learning models. In addition to the parameters related to data, backbone network, and optimizer that are common to all deep learning methods, EXCOST introduces a total of 5 parameters. These are $w_u$ (weight for CIL), $\delta$ (margin), and three scheduling functions: $\Psi$ (threshold for CIL), $\Phi_C$ (labeling rate based on confidence), and $\Phi_S$ (labeling rate based on exemplar similarity).
>
> Let's also consider the parameter settings of other semi-supervised learning models: FixMatch [1] has 4 parameters: decay of EMA, $\lambda_u$ (unlabeled loss weight), $T$ (temperature), and $\tau$ (confidence threshold); MixMatch [2] has 5 parameters: decay of EMA, $\lambda_u$ (unsupervised loss weight), $\alpha$ (parameter for MixUp), $T$ (temperature), and $K$ (number of augmentations). Therefore, in terms of quantity, EXCOST does not involve an excessive number of hyperparameters.
>
> For $\delta$ and $\Psi$, we use the same settings across all datasets, without the need for adjustments. Due to inherent differences in datasets and the lack of feedback information on the learning status in our algorithm, we need to set the labeling rate scheduling functions ($\Phi_C$ and $\Phi_S$) based on the dataset. This is similar to how humans exhibit variations in learning across different tasks. In future work, we aim to address this issue by incorporating active learning or reinforcement learning to mimic human-like learning in classification tasks.
>
> It is worth noting that for two significantly different datasets, MNIST and CIFAR-10, even when employing completely identical settings for the 5 mentioned hyperparameters, EXCOST consistently achieves excellent results. Furthermore, our results, except for CIFAR-100, are obtained using the Adam [3] optimization algorithm, rather than meticulously tuning parameters with the SGD optimizer.

---

> ### Author Response · Authors · 2023-11-17
> **Response to Reviewer GnDx (2/2)**
>
> > While the paper mentions that the computational burden is reduced by deferring the computation of exemplar feature vectors until the labeling phase, it lacks a comprehensive analysis of the algorithm's computational efficiency. The practical implications and potential computational overhead of the entire algorithm should be subject to in-depth investigation and evaluation.
>
> In the context of semi-supervised learning, the number of unlabeled samples is typically much larger than the number of labeled samples. Due to the fact that EXCOST saves the feature vectors of all samples during training, there is no need to compute them again during the labeling phase. As a result, the entire labeling phase of EXCOST can usually be completed within seconds.
>
> For evaluating the learning efficiency of deep learning models, it is common to compare the training times of different models on the same hardware platform. However, obtaining such data can be challenging as many related papers do not provide it, and experiments are often conducted on different hardware platforms. Therefore, in order to carry out this evaluation, it would be necessary to test all baseline models on the same platform, which would require a significant amount of computing resources. Unfortunately, we only have access to four RTX 3090 GPUs, making it difficult to complete this evaluation within a short period of time. However, we can provide some reference data using the experimental setup described in Section 4.1 of our paper. We calculated the average number of minutes consumed by a single random seed for each experiment using a single RTX 3090 GPU, as shown in the table below:
>
> | MNIST@20 | MNIST@50 | MNIST@100 | CIFAR-10@250 | CIFAR-10@4000 | CIFAR-100@2500 | CIFAR-100@10000 | SVHN@250 | SVHN@1000 |
> | -------- | -------- | --------- | ------------ | ------------- | -------------- | --------------- | -------- | --------- |
> | 81       | 82       | 82        | 1457         | 1484          | 4556           | 4656            | 746      | 737       |
>
> It is worth noting that the reported time consumption is also influenced by the experimental conditions at that time (such as the temperature of the GPU), and there may be slight performance variations among the four GPUs. To make a comparison, we examined the original papers of all baseline models. Unfortunately, we did not find relevant information in these papers.
>
>
> If we have missed any key points or if you have any further questions, we would be happy to continue the discussion.
>
>
> ## Reference
>
> [1] Kihyuk Sohn, David Berthelot, Nicholas Carlini, Zizhao Zhang, Han Zhang, Colin A Raffel, Ekin Dogus Cubuk, Alexey Kurakin, and Chun-Liang Li. FixMatch: Simplifying semi-supervised learning with consistency and confidence. In Advances in Neural Information Processing Systems, volume 33, pp. 596–608. Curran Associates, Inc., 2020.
>
> [2] David Berthelot, Nicholas Carlini, Ian Goodfellow, Nicolas Papernot, Avital Oliver, and Colin A Raffel. MixMatch: A holistic approach to semi-supervised learning. In Advances in Neural Information Processing Systems, volume 32. Curran Associates, Inc., 2019.
>
> [3] Diederik P. Kingma and Jimmy Ba. Adam: A method for stochastic optimization, 2017.

---

### Meta-Review · Area_Chair_WHqs · 2023-12-05

**Metareview:**

The paper introduces the Exemplar-Contrastive Self-Training (EXCOST) algorithm for semi-supervised learning, along with a novel regularization term, Category-Invariant Loss (CIL). The paper aims to improve the reliability of pseudo-labels for unlabeled data and to ensure consistency of class probabilities under different perturbations. It demonstrates strong performance on various benchmark datasets. While the reviewers acknowledge the paper is well written, with an interesting approach, and thorough experiments. However, the reviewers also point out several limitations of the work: such as lack of comprehensive computational analysis, the lack of comparative advantage compared to existing approaches. There are also some questions around experimental details, and baselines. Overall, reviewers do not advocate for acceptance, as the work still needs further refinement to be ready to publish.

**Justification For Why Not Higher Score:**

No support opinions from reviewers

**Justification For Why Not Lower Score:**

this is lowest

---

### Decision · Program_Chairs · 2024-01-16

Reject